

# Lacunes and type 2 diabetes mellitus have a joint effect on cognitive impairment: a retrospective study

Hong Zhou*, Jun Hu*, Peihan Xie, Yulan Dong, Wen Chen, Huiting Wu, Yihong Jiang, Hao Lei, Guanghua Luo and Jincai Liu

Department of Radiology, First Affiliated Hospital of University of South China, Hengyang, China
* These authors contributed equally to this work.

Corresponding authors
Guanghua Luo,
1579839814@qq.com
Jincai Liu, liujincai6353@163.com

## ABSTRACT

**Objective:** To evaluate the joint effects of cerebral small vessel disease (CSVD)-related imaging biomarkers in patients of type 2 diabetes mellitus (T2DM) with cognitive impairment.

**Methods:** This study is a retrospective cohort study. A total of 227 participants (115 patients with T2DM and 112 healthy control subjects) were enrolled in this study. Cognitive function assessments were evaluated using the Mini-Mental State Examination and the Montreal Cognitive Assessment. The burden of CSVD markers, including the lacunes, white matter hyperintensities (WMH), cerebral microbleeds (CMBs), and enlarged perivascular spaces (PVS), was identified by magnetic resonance imaging and evaluated using small vessel disease (SVD) scores (0–4). The subjects were divided into two groups based on the results of the cognitive function assessments. The synergy index was used to estimate the biological interactions between T2DM and lacunes.

**Results:** There was a significant correlation between T2DM and cognitive impairment ($p < 0.001$, $\chi^2$ test). In patients with diabetes, cognitive impairment was significantly associated with both the presence of lacunes ($p < 0.01$, $\chi^2$ test) and increased total SVD burden scores ($p < 0.01$, $\chi^2$ test). Regarding CMBs, only the existence of lobar CMBs was correlated with cognitive impairment ($p < 0.05$, $\chi^2$ test). The joint effect tended to be larger than the independent effects of T2DM and lacunes on cognitive impairment (adjusted odds ratio [OR]: 7.084, 95% CI [2.836–17.698]; synergy index: 10.018, 95% CI [0.344–291.414]).

**Conclusions:** T2DM and the presence of lacunes are significantly correlated with cognitive impairment. There was a joint effect of T2DM and lacunes on cognitive impairment.

## INTRODUCTION

Type 2 diabetes mellitus (T2DM) has a very high prevalence worldwide, and the prevalence is increasing year by year. Patients with diabetes often have multiple complications, and cognitive impairment is one of the most common and disruptive

complications (*Gorelick et al., 2011*; *Geijselaers et al., 2015*; *Harreiter & Roden, 2019*). Cognitive dysfunction not only aggravates deterioration of a patient's physical condition but also causes economic harm (*Moon et al., 2021*). To date, the relationships between diabetic complications and specific neuropathological causes of cognitive impairment are not fully understood. However, some factors potentially involved in these pathophysiological changes include insulin resistance, high fasting blood glucose, endothelial dysfunction, hypercoagulation, hypertension and comorbid abdominal obesity.

Many epidemiological studies have demonstrated possible associations between diabetes and cerebral small vessel disease (CSVD), in particular with lacunar stroke (*Air & Kissela, 2007*; *Sorensen et al., 2016*; *Del Brutto et al., 2018*; *Liu et al., 2018*). The common brain imaging manifestations of diabetes are cerebral infarction and lacunes (*Liu et al., 2018*; *Passiak et al., 2019*). Previous studies have shown that diabetes significantly increases the incidence of lacunes. The presence of lacunes has also been associated with reduced cognitive assessment scores (global end-of-life cognition and animal category naming fluency) in patients with diabetes (*Abner et al., 2016*). It seems that arteriolar perfusion dysfunction can be affected by long-term ischemia and oxidative stress in diabetics (*Fernando et al., 2006*; *Sorensen et al., 2016*). Abnormal cerebral blood perfusion can lead to hypoxia in brain tissue through decreased cerebral cortex metabolism and white matter damage, and therefore, diabetic patients often experience cognitive decline (*Wallin et al., 2018*). The above studies suggest that the existence of CSVD may contribute to cognitive impairment in patients with diabetes.

Magnetic resonance imaging (MRI) is one of the noninvasive imaging modalities for detecting CSVD and it has been standardized to describe various imaging features of CSVD (*Wallin et al., 2018*). According to international neuroimaging standards, total SVD scores consist of four parts: lacunes, white matter hyperintensities (WMH), cerebral microbleeds (CMBs), and enlarged perivascular spaces (PVS) (*Wardlaw et al., 2013*). All of these biomarkers have been individually linked to some cerebral dysfunction, especially cognitive impairment, but some clinical indicators of CSVD-related markers, such as WMH, CMBs, and lacunes, show poor correlations with neuropsychological performance (*Passiak et al., 2019*). Accumulated research agrees on the definition of lacunes as a sequela of acute subcortical infarction and that deep cerebral hemorrhages can also form lacunes (*Low et al., 2019*). Exploring these imaging markers is helpful for studying diabetes with cognitive impairment.

According to previous studies, the above four neuroimaging biomarkers and their weighted total SVD scores have well-validated associations with vascular risk factors (*Yakushiji et al., 2014*; *Hara et al., 2019*). Therefore, we speculate that the method of quantifying CSVD-related imaging markers can help study the cognitive impairment of patients with diabetes. This study aims to evaluate the joint effects of CSVD-related imaging biomarkers in T2DM with cognitive impairment and investigate whether the SVD score can be used to predict the occurrence of cognitive impairment in diabetic participants.

## MATERIALS AND METHODS

### Subjects

Participants enrolled in the program had to be able to independently complete brain health check tests and undergo MRI scanning for further evaluation. Participants also had to voluntarily accept the regulations provided on the informed consent form. Inclusion criteria: (1) diagnosis criteria for diabetes: Fasting plasma glucose (FPG) ≥126 mg/dl (7.0 mmol/l), or 2-h values in the oral glucose tolerance test (OGTT) ≥200 mg/dl (11.1 mmol/l), or glycated haemoglobin (HbA1c) ≥6.5%. Exclude other types of diabetes such as type 1 diabetes and gestational diabetes based on clinical diagnosis (*Harreiter & Roden, 2019*); (2) were over 18 years old and (3) could cooperate and complete the Mini-Mental State Examination, version 2 (MMSE-2) and Montreal Cognitive Assessment (MoCA) conventional tests. The exclusion criteria were as follows: (1) patients had severe brain organic diseases and other serious physical diseases; (2) women were lactating/pregnant; (3) patients had magnetic resonance contraindications; (4) patients had a family history of dementia; (5) patients refused to sign the informed consent form; or (6) MRI images did not validate the diagnostic criteria.

All 479 participants underwent a 3.0-T MRI scan between May 2017 and January 2019. Individual clinical characteristics, such as age, sex, hyperlipidemia, smoking status, presence of hypertension and diabetes, and years of schooling completed were collected, and the MMSE-2 and the MoCA were completed. Neuropsychological questionnaires were assessed by a professionally trained nurse in the cognitive function ward. MRI studies were evaluated by two radiologists (3 and 9 years of experience in neuroimaging), and controversial MRI images were subsequently evaluated by more experienced radiologists. The two radiologists were blinded to all the clinical information and independently evaluated the characteristics of each patient's brain.

After excluding subjects who had missing or unsatisfactory MRI data for analysis ($n = 50$), missing basic clinical information ($n = 17$), or missing cognitive assessments ($n = 185$), 227 subjects were included in this study (Fig. 1). This study was approved by the human ethics review board of the University of South China. All participants provided both oral and written informed consent. The project was registered at the Chinese Clinical Trial Registry (ChiCTR1900027781).

### Baseline assessment

According to current diabetes classifications (*Harreiter & Roden, 2019*), the definition of T2DM is a fasting serum glucose level ≥126 mg/dl or HbA1c ≥6.5%. Hypertension is defined as systolic blood pressure (SBP) ≥140 mmHg and/or diastolic blood pressure (DBP) ≥90 mmHg. We systematically collected education information (years of schooling) and smoking history. The MMSE-2 scale defines cognitive impairment as a score ≤17 points for uneducated participants, ≤20 points for patients with an education period of less than or equal to 6 years, and ≤24 points for patients with an education period of more than 6 years (*Cockrell & Folstein, 1988*). Patients with a total score of <26 points on the

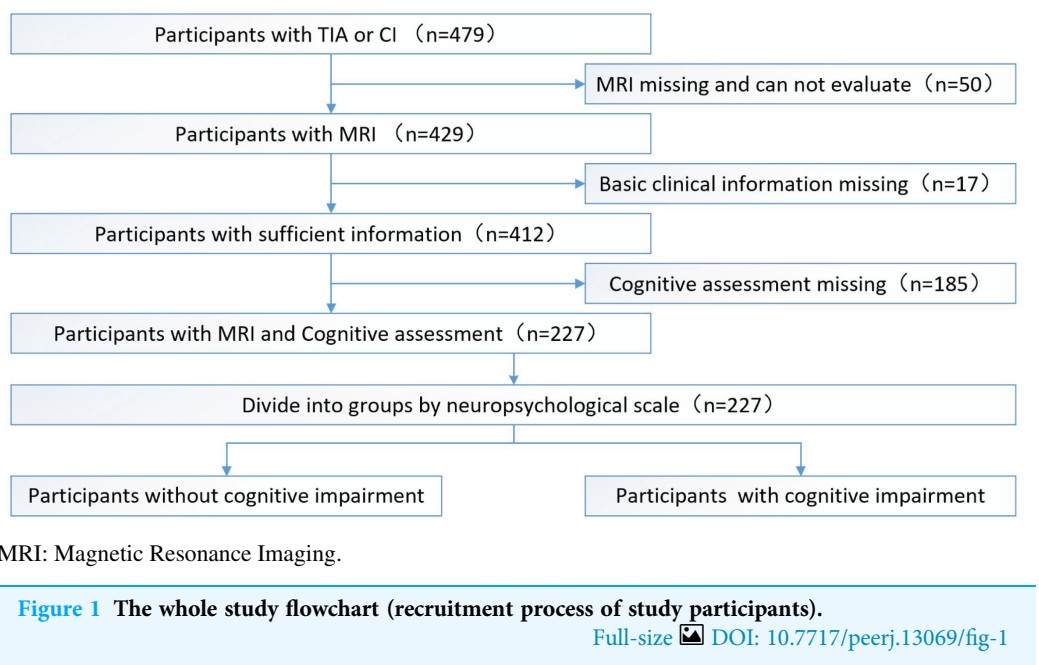

MRI: Magnetic Resonance Imaging.

**Figure 1 The whole study flowchart (recruitment process of study participants).**

MoCA scale are considered to have cognitive impairment (with an additional point added if the subject has been educated for less than or equal to 12 years) (*Nasreddine et al., 2005*).

## Brain MRI acquisition

Brain MRI scans were obtained at The First Affiliated Hospital of University of South China using the Philips Achieva 3.0-T system with 8-channel SENSE receiver. The main sequences included the following. T1-weighted imaging: repetition time (TR), 600 ms; echo time (TE), 10 ms; flip angle (FA), 70°; section thickness, 5 mm; gap width, 1 mm; matrix, $256 \times 163$ mm$^2$; and field of view (FOV), $230 \times 230$ mm$^2$. Fast spin-echo T2-weighted imaging: TR, 3,000 ms; TE, 80 ms; FA, 90°; section thickness, 5 mm; gap width, 1 mm; matrix, $400 \times 255$ mm$^2$; and FOV, $230 \times 230$ mm$^2$. Fluid-attenuated inversion recovery (FLAIR) imaging: TR, 11,000 ms; inversion time, 2,800 ms; TE, 120 ms; FA, 120°; section thickness, 4 mm; gap width, 1 mm; matrix, $232 \times 101$ mm$^2$; and FOV, $230 \times 230$ mm$^2$. Susceptibility-weighted imaging (SWI): TR, 18 ms; TE, 26 ms; FA, 10°; section thickness, 0.6 mm; gap width, 1 mm; matrix, $244 \times 200$ mm$^2$; and FOV, $220 \times 220$ mm$^2$.

## Lacunes, WMH, CMBs, and PVS analysis

All evaluators of the MRI examinations were blinded to the clinical data. The MR images were evaluated using the prestudy cerebral SVD study standard (*Kim, MacFall & Payne, 2008*; *Gregoire et al., 2009*; *Wardlaw et al., 2013*). Lacunes were defined as focal or ovoid, 3 to 15 mm in diameter, fluid-filled signal cavities located in the area of a perforating arteriole. On FLAIR images, lacunes generally have a central hypointensity, with similar signals as cerebrospinal fluid (CSF), surrounded by hyperintensity (*Wardlaw et al., 2013*). WMH were defined as hyperintense areas in the T2-weighted image and void-free hyperintense areas on FLAIR sequences. We graded the periventricular and deep lesions

according to the modified Fazekas scale. Periventricular hyperintensities were graded as absent (Grade 0), caps or pencil-thin lining (Grade 1), bands (Grade 2), or irregular extension into the deep white matter (Grade 3), and the deep lesions were graded as absent (Grade 0), punctuate (Grade 1), beginning confluence (Grade 2), or confluent (Grade 3) (*Fazekas et al., 1987*). We counted the total number of CMBs in the deep, lobar, and infratentorial regions except for lesions in the subarachnoid space and low symmetry signal areas of the globus pallidus (as these possibly represent adjacent pial vessels and calcification, respectively) (*Gregoire et al., 2009*). The number of CMBs was graded as absent, mild (1–2), moderate (3–10), or severe (>10) (*Lee et al., 2002*). PVS was defined as small (<3 mm) structures with the same signal as CSF that followed the area of perforating arteries. The severity of PVS was assessed in the basal ganglia (BG), centrum semiovale, and hippocampus (Hip). In the BG, PVS was graded in the slice that had the largest number of PVS and was scored as follows: the number of PVS <5 (Grade 1), 5–10 (Grade 2), >10 but still countable (Grade 3), or uncountable (Grade 4). In the centrum semiovale, PVS was scored as follows: the sum of the number of PVS in all slices <10 (Grade 1), the sum of the number of PVS in all slices >10 but the number of PVS <10 in any slice (Grade 2), the number of PVS was 10–20 in the slice containing the largest number of PVS (Grade 3), or the number of PVS >20 in the slice containing the largest number of PVS (Grade 4) (*Doubal et al., 2010*). In the Hip, the number of PVS in the total slices of the Hip and parahippocampal gyrus was scored. PVS were scored as follows: number of PVS <5 (Grade 1), 5–10 (Grade 2), or >10 (Grade 3) (*Zhang et al., 2014*).

## SVD score analysis

In accordance with previous studies (*Klarenbeek et al., 2013*; *Staals et al., 2014*), if the number of lacunes or CMBs was greater than or equal to one, they were each recorded as 1 point. The presence of white matter hyperintensities was defined as periventricular WMH Fazekas score 3 and/or deep WMH Fazekas score 3 and recorded as 1 point. The presence of moderate to severe PVS was recorded as 1 point. The total SVD score consists of the above four parts and could range from 0 to 4. The subjects were further divided based on the total SVD score: a none to mild group (SVD score: 0–1) and a moderate to severe group (SVD score: 2–4) (*Yakushiji et al., 2014*; *Hara et al., 2019*).

## Statistical analysis

According to the above baseline assessment criteria, all subjects were divided into two groups (cognitive impairment group and noncognitive impairment group) using the MMSE-2 and MoCA scale. To apply the research to diabetic participants, we selected the appropriate assessment using chi-square or Fisher's exact tests. The subjects were divided into two groups, and all baseline and imaging data were analyzed. Categorical and ordinal variables are presented as $n$ (%) and were analyzed with chi-square or Fisher's exact tests. Continuous variables are presented as the mean ± standard deviation (SD). The normality of data distributions and homogeneity of continuous variables were examined before analysis, and then the appropriate $t$-test was used. The joint effect of diabetes and lacunes was evaluated by calculating the synergy index using the method

**Table 1 General characteristics of the study population.** A statistical analysis of age found that when using the MMSE-2 assessment, the non-cognitive group had a normal distribution and the cognitively disabled group did not follow the normal distribution. Therefore, a non-parametric test was selected. When using the MMSE-2 assessment, the non-cognitive disorder group had a normal distribution, and the cognitively impaired group did not follow the normal distribution. Therefore, a non-parametric test was selected. Chi-square tests were performed on sex, education, diabetes, smoking, hypertension, and hyperlipidemia. Among the MOCA assessment groups, the number of individual tests for diabetes and gender was less than 5, so the continuous correction value test was used, and the others were routinely selected by Pearson's chi-squared test. There were statistical differences in the MMSE-2 assessment in the diabetic group, so the MMSE-2 was selected as a scale for cognitive assessment before subsequent analysis.

| | Entire cohort (n = 227) | Evaluated by using MMSE-2 | | | Evaluated by using MOCA | | |
|---|---|---|---|---|---|---|---|
| | | Without CI (n = 155) | With CI (n = 72) | p | Without CI (n = 16) | With CI (n = 211) | p |
| Sex (men, %) | 146 (64.3%) | 107 (69.0%) | 39 (54.2%) | 0.030 | 15 (93.8%) | 131 (62.1%) | 0.023 |
| Age (median) | 62.59 ± 10.84 | 61.65 ± 11.00 | 64.64 ± 10.27 | 0.053 | 59.06 ± 12.12 | 62.86 ± 10.72 | 0.211 |
| Education (PSE, %) | 90 (39.6%) | 58 (37.4%) | 32 (44.4%) | 0.314 | 10 (62.5%) | 80 (37.9%) | 0.053 |
| Diabetes (%) | 115 (51.7%) | 60 (38.7%) | 55 (76.4%) | <0.001 | 4 (25.0%) | 111 (52.6%) | 0.061 |
| Smoke | 64 (28.2%) | 49 (31.6%) | 15 (20.8%) | 0.093 | 6 (37.5%) | 58 (27.5%) | 0.396 |
| Hyperlipidemia | 74 (32.6%) | 46 (29.7%) | 28 (38.9%) | 0.168 | 5 (31.3%) | 69 (32.7%) | 0.905 |
| Hypertension (%) | 169 (74.4%) | 109 (70.3%) | 60 (83.3%) | 0.036 | 8 (50.0%) | 161 (76.3%) | 0.108 |

**Note:**
PSE: post-secondary education. MMSE: Mini-mental State Examination. MOCA: Montreal Cognitive Assessment. CI: cognitive impairment.

introduced by *Andersson et al. (2005)*. An Excel sheet was built to calculate the synergy index. The Excel formula is Synergy index = $[RR_{11} – 1]/[(RR_{10} – 1) + (RR_{01} – 1)]$. In this formula, $RR_{11}$ is the relative risk of both risk factors being present, $RR_{10}$ is the relative risk of the first risk factor being absent and the second risk factor being present, and $RR_{01}$ is the relative risk of the second risk factor being absent and the first risk factor being present. If there was no biological interaction, the synergy index is equal to 1. If the synergy index is larger than 1, it indicates that there is a synergistic effect of the two factors. Otherwise, there is only a joint effect of the two factors. All statistical analyses were performed with SPSS software (Version 25; SPSS, Chicago, IL, USA), the Statistical Analysis System (Version 9.4; SAS Institute, Cary, NC, USA) and Microsoft Excel (Version 2019; Microsoft Corp, Redmond, WA, USA). Statistical significance was established at $p < 0.05$ for all tests.

# RESULTS

## Participant characteristics

Cohen's kappa values were calculated as a measure of interrater reliability. The Cohen's kappa values for lacunes, WMH, CMBs, and PVS were 0.76, 0.84, 0.89, and 0.66, respectively. The data were evaluated by two imaging specialists who were unaware of the results of each other's assessment during the assessment period. Table 1 shows the demographic and clinical findings for the entire cohort (n = 227: 64.3% male; mean age: 62.5 years). Among the entire cohort, there were 115 (51.7%) people with diabetes, 74 (32.6%) people with hyperlipidemia, and 169 (74.4%) people with hypertension. There were 90 (39.6%) people with postsecondary education among the entire cohort and 64 (28.2%) people with a history of smoking.

**Table 2 Small blood vessel scores using MMSE-2 scoring in the entire cohort.** Statistical tests showed that the data all conformed to the normal distribution with or without cognitive impairment, and then the homogeneity of variance was further tested. ANOVA analysis was used to conclude that the total CMBs score, total lacunes score, lobar CMBs score, total SVD burden score and total SVD burden score ≥3 was heterogeneity of variance, therefore, Welch's adjusted unpaired $t$-test was selected. For others, an unpaired $t$-test was selected.

| | Evaluated by using MMSE-2 | | | | |
| --- | --- | --- | --- | --- | --- |
| | **Without CI** | **With CI** | **F** | **t** | **p** |
| Total WMH score | 0.39 ± 0.490 | 0.51 ± 0.503 | 3.247 | −1.707 | 0.089 |
| Periventricular WMH | 1.43 ± 0.925 | 1.64 ± 0.954 | 3.247 | −1.599 | 0.111 |
| Deep WMH | 0.39 ± 0.490 | 0.51 ± 0.503 | 0.497 | −1.707 | 0.093 |
| Total CMBs score | 0.60 ± 0.491 | 0.71 ± 0.458 | 12.128 | −1.579 | 0.116 |
| Lobar CMBs | 0.31 ± 0.464 | 0.43 ± 0.499 | 8.485 | −1.737 | 0.085 |
| Deep CMBs | 0.48 ± 0.501 | 0.60 ± 0.494 | 4.822 | −1.693 | 0.094 |
| Subtentorial CMBs | 0.26 ± 0.439 | 0.26 ± 0.444 | 0.034 | −0.093 | 0.926 |
| Total Lacunes score | 0.45 ± 0.499 | 0.68 ± 0.470 | 15.489 | −3.446 | ≤0.001 |
| Total perivascular spaces score | 1.00 ± 0.000 | 1.00 ± 0.000 | | | |
| Basal ganglia PVS | 3.11 ± 0.778 | 3.22 ± 0.859 | 3.065 | −0.981 | 0.328 |
| Centrum semiovale PVS | 3.46 ± 0.647 | 3.54 ± 0.670 | 0.030 | −0.896 | 0.371 |
| Hippocampus PVS | 2.32 ± 0.728 | 2.50 ± 0.732 | 0.041 | −1.769 | 0.078 |
| Total SVD burden score | 2.42 ± 1.167 | 2.90 ± 1.009 | 10.248 | −3.192 | <0.01 |
| ≥2 | 3.02 ± 0.850 | 3.17 ± 0.752 | 1.805 | −1.211 | 0.228 |
| ≥3 | 3.56 ± 0.499 | 3.48 ± 0.505 | 0.608 | 0.900 | 0.370 |

**Note:**
WMH: white matter hyperintensities. CMBs: cerebral microbleeds. PVS: perivascular spaces. SVD: small vessel disease. MMSE: Mini-mental State Examination. CI: cognitive impairment.

## Markers of SVD and cognitive assessment outcomes

The numbers of subjects with SVD scores of 0, 1, 2, 3, and 4 were 0 (0%), 55 (24.2%), 51 (22.4%), 57 (25.8%), and 64 (28.2%), respectively. Table 1 shows MMSE-2 scores for the entire cohort. There was a significant correlation between diabetes and cognitive impairment ($p < 0.001$, $\chi^2$ test). The entire cohort was evaluated based on MMSE-2 scores, and Table 2 shows that total SVD scores were higher in the cognitive impairment group than in the noncognitive impairment group ($p < 0.01$, $\chi^2$ test) and that the existence of lacunes was significantly associated with cognitive impairment ($p \leq 0.001$, $\chi^2$ test). Table 3 shows the statistics for diabetes patients in the entire cohort that show that increased age ($p < 0.05$, $\chi^2$ test) was associated with cognitive impairment. Table 4 shows no statistically significant differences among the general characteristics in the non-diabetic group. Table 5 shows that in patients with diabetes, cognitive impairment was significantly associated with the presence of lacunes ($p < 0.01$, $\chi^2$ test) and higher total SVD burden scores ($p < 0.01$, $\chi^2$ test). Regarding CMBs, only the existence of lobar CMBs was related to cognitive impairment ($p < 0.05$, $\chi^2$ test).

## Joint effect of diabetes and lacunes on cognitive impairment

Table 6 shows that although there were no significant synergistic effects of diabetes and lacunes on cognitive impairment, the joint effects tended to be larger than independent

**Table 3 General characteristics of diabetic patients in this cohort.** Statistical analysis of age revealed that both groups conformed to a normal distribution and the variance tests were not homogeneous. Therefore, Welch's adjusted unpaired $t$-test was selected. For other factors, a chi-square test was selected.

| | Entire cohort (n = 115) | Evaluated by using MMSE-2 | | | |
| --- | --- | --- | --- | --- | --- |
| | | DM Without CI (n = 60) | DM With CI (n = 55) | $\chi^2/t$ | $p$ |
| Sex (men, %) | 65 (56.5%) | 35 (58.3%) | 30 (54.5%) | 0.168 | 0.682 |
| Age (median) | 62.42 ± 9.382 | 62.52 ± 9.038 | 66.49 ± 9.390 | −2.312 | <0.05 |
| Education (PSE, %) | 39 (33.9%) | 17 (28.3%) | 22 (40.0%) | 1.743 | 0.187 |
| Smoke | 29 (25.2%) | 17 (28.3%) | 12 (21.8%) | 0.646 | 0.422 |
| Hyperlipidemia | 36 (31.3%) | 16 (26.7%) | 20 (36.4%) | 1.255 | 0.263 |
| Hypertension (%) | 96 (83.5%) | 49 (81.7%) | 47 (85.5%) | 0.299 | 0.585 |

Note:
PSE: post-secondary education. MMSE: Mini-mental State Examination. CI: cognitive impairment. DM: diabetes mellitus

**Table 4 General characteristics of non-diabetic patients in this cohort.** Statistical analysis of age revealed that both groups conformed to a normal distribution and the variance tests were not homogeneous. Therefore, Welch's adjusted unpaired $t$-test was selected. For other factors, a chi-square test was selected.

| | Entire cohort (n = 112) | Evaluated by using MMSE-2 | | | |
| --- | --- | --- | --- | --- | --- |
| | | Without CI (n = 95) | With CI (n = 17) | $\chi^2/t$ | $p$ |
| Sex (men, %) | 81 (72.3%) | 72 (75.8%) | 9 (52.9%) | 3.761 | 0.076 |
| Age (median) | 60.72 ± 11.907 | 61.09 ± 12.083 | 58.65 ± 10.977 | 0.779 | 0.438 |
| Education (PSE, %) | 51 (45.5%) | 41 (43.2%) | 10 (58.8%) | 1.427 | 0.232 |
| Smoke | 35 (31.3%) | 32 (33.7%) | 3 (17.6%) | 1.726 | 0.189 |
| Hyperlipidemia | 38 (33.9%) | 30 (31.6%) | 8 (47.1%) | 1.541 | 0.214 |
| Hypertension (%) | 73 (65.2%) | 60 (63.2%) | 13 (81.5%) | 1.126 | 0.289 |

Note:
PSE: post-secondary education. MMSE: Mini-mental State Examination. CI: cognitive impairment.

**Table 5 Small blood vessel scores using MMSE-2 scoring in diabetic.** All data were statistically tested for normal distribution. ANOVA analysis was used to test the homogeneity of variance and found that the total Lacunes score, lobar CMBs score, total SVD burden score and total SVD burden score ≥3 was heterogeneity of variance. Therefore, Welch's adjusted unpaired $t$-test was selected. For others, an unpaired $t$-test was selected.

| | Evaluated by using MMSE-2 | | | | |
| --- | --- | --- | --- | --- | --- |
| | Without CI | With CI | F | $t$ | $p$ |
| Total WMH score | 0.50 ± 0.504 | 0.58 ± 0.498 | 1.622 | −0.875 | 0.384 |
| Periventricular WMH | 1.58 ± 0.962 | 1.76 ± 0.922 | 0.285 | −1.024 | 0.308 |
| Deep WMH | 0.50 ± 0.504 | 0.58 ± 0.498 | 1.622 | −0.875 | 0.384 |
| Total CMBs score | 0.73 ± 0.446 | 0.82 ± 0.389 | 4.840 | −1.089 | 0.278 |
| Lobar CMBs | 0.30 ± 0.462 | 0.51 ± 0.505 | 5.340 | −2.311 | <0.05 |
| Deep CMBs | 0.63 ± 0.486 | 0.69 ± 0.466 | 1.672 | −0.647 | 0.519 |
| subtentorial CMBs | 0.27 ± 0.446 | 0.25 ± 0.440 | 0.086 | 0.147 | 0.884 |
| Total Lacunes score | 0.52 ± 0.504 | 0.78 ± 0.417 | 27.107 | −3.059 | <0.01 |

| | Evaluated by using MMSE-2 | | | | |
| --- | --- | --- | --- | --- | --- |
| | Without CI | With CI | F | t | p |
| Total perivascular spaces score | 1.00 ± 0.000 | 1.00 ± 0.000 | | | |
| Basal ganglia PVS | 3.18 ± 0.676 | 3.33 ± 0.771 | 4.130 | −1.060 | 0.291 |
| Centrum semiovale PVS | 3.52 ± 0.624 | 3.62 ± 0.552 | 3.109 | −1.251 | 0.214 |
| Hippocampus PVS | 2.40 ± 0.669 | 2.62 ± 0.593 | 2.605 | −1.844 | 0.068 |
| Total SVD burden score | 2.68 ± 1.097 | 3.18 ± 0.884 | 7.508 | −2.693 | <0.01 |
| ≥2 | 3.02 ± 0.869 | 3.31 ± 0.729 | 1.890 | −1.815 | 0.073 |
| ≥3 | 3.59 ± 0.499 | 3.55 ± 0.504 | 0.686 | 0.414 | 0.680 |

**Note:**
WMH: white matter hyperintensities. CMBs: cerebral microbleeds. PVS: perivascular spaces. SVD: small vessel disease. MMSE: Mini-mental State Examination. CI: cognitive impairment.

**Table 6 Joint effect of diabetes and lacunas to cognitive impairment.**

| Factor 1 | Factor 2 | Without cognitive impairment | Cognitive impairment | | |
| --- | --- | --- | --- | --- | --- |
| | | N | N | Adjusted OR | (95% CI) |
| Diabetes | Lacuna | | | | |
| (−) | (−) | 57 | 11 | 1.00 | [ref.] |
| (−) | (+) | 38 | 6 | 0.818[a] | [0.279–2.400] |
| (+) | (−) | 29 | 12 | 2.144[a] | [0.844–5.477] |
| (+) | (+) | 31 | 43 | 7.188[a] | [3.250–15.895] |
| | | Synergy index | | 6.429[a] | [0.753–54.911] |
| (−) | (−) | 57 | 11 | 1.00 | [ref.] |
| (−) | (+) | 38 | 6 | 0.674[b] | [0.218–2.081] |
| (+) | (−) | 29 | 12 | 1.905[b] | [0.734–4.944] |
| (+) | (+) | 31 | 43 | 6.248[b] | [2.743–14.230] |
| | | Synergy index | | 9.059[b] | [0.316–259.394] |
| (−) | (−) | 57 | 11 | 1.00 | [ref.] |
| (−) | (+) | 38 | 6 | 0.730[c] | [0.234–2.273] |
| (+) | (−) | 29 | 12 | 1.779[c] | [0.679–4.663] |
| (+) | (+) | 31 | 43 | 6.221[c] | [2.719–14.235] |
| | | Synergy index | | 10.256[c] | [0.236–445.414] |
| (−) | (−) | 57 | 11 | 1.00 | [ref.] |
| (−) | (+) | 38 | 6 | 0.754[d] | [0.242–2.353] |
| (+) | (−) | 29 | 12 | 1.807[d] | [0.688–4.746] |
| (+) | (+) | 31 | 43 | 6.411[d] | [2.785–14.755] |
| | | Synergy index | | 9.647[d] | [0.302–308.531] |
| (−) | (−) | 57 | 11 | 1.00 | [ref.] |
| (−) | (+) | 38 | 6 | 0.705[e] | [0.222–2.241] |
| (+) | (−) | 29 | 12 | 1.959[e] | [0.735–5.221] |
| (+) | (+) | 31 | 43 | 7.179[e] | [3.038–16.965] |
| | | Synergy index | | 9.301[e] | [0.442–195.935] |

(Continued)

| Factor 1 | Factor 2 | Without cognitive impairment | Cognitive impairment | | |
|---|---|---|---|---|---|
| | | N | N | Adjusted OR | (95% CI) |
| (−) | (−) | 57 | 11 | 1.00 | [ref.] |
| (−) | (+) | 38 | 6 | 0.684[f] | [0.212–2.205] |
| (+) | (−) | 29 | 12 | 2.104[f] | [0.784–5.648] |
| (+) | (+) | 31 | 43 | 7.179[f] | [2.996–17.202] |
| | | Synergy index | | 7.839[f] | [0.547–112.447] |
| (−) | (−) | 57 | 11 | 1.00 | [ref.] |
| (−) | (+) | 38 | 6 | 0.629[g] | [0.190–2.078] |
| (+) | (−) | 29 | 12 | 1.978[g] | [0.724–5.405] |
| (+) | (+) | 31 | 43 | 7.084[g] | [2.836–17.698] |
| | | Synergy index | | 10.018[g] | [0.344–291.414] |

**Notes:**
CI, confidence interval; OR, odds ratio; ref, reference.
[a] Without any adjusted.
[b] Adjusted for age.
[c] Adjusted for sex and age.
[d] Adjusted for sex, age, and history of smoking.
[e] Adjusted for age, sex, history of smoking, and education.
[f] Adjusted for sex, age, smoking status, education, and hyperlipidemia.
[g] Adjusted for sex, age, smoking status, education, hyperlipidemia, and hypertension.

effects after adjusting for sex, age, smoking status, education, hyperlipidemia and hypertension (adjusted OR: 7.084, 95% CI [2.836–17.698]; synergy index: 10.018, 95% CI [0.344–291.414]).

# DISCUSSION

In this study, we used MRI to demonstrate changes in small blood vessel damage in diabetic patients with cognitive impairment and to evaluate SVD scores in patients with diabetes with cognitive impairment. Our study showed that the presence of lacunes was significantly connected with cognitive impairment in diabetic patients and that the presence of diabetes and lacunes had a joint effect on the risk of cognitive impairment in addition to increased total SVD burden scores and worse cognitive performance. We have reason to believe that the presence of lacunes is an important imaging biomarker in diabetic patients with cognitive impairment. The presence of lacunes can be used as important evidence for assessing cognitive impairment in patients with diabetes because the appearance of lacunes is significantly linked with the decline in cognitive ability in this population. This study provided further evidence that monitoring lacunes in diabetic patients can help slow the progression of cognitive impairment.

In patients with diabetes, many factors may be involved in these pathophysiological changes, such as insulin resistance, high fasting blood glucose, endothelial dysfunction, and hypercoagulation (*Air & Kissela, 2007*; *Sorensen et al., 2016*; *Liu et al., 2018*). Arteriolar perfusion dysfunction may be affected by long-term ischemia and oxidative stress (*Fernando et al., 2006*; *Sorensen et al., 2016*). Disorders of the blood vessel walls and hyaline accumulation can cause blockage of the arteries, which can lead to the formation of

lacunes (*Fisher, 1982*). For example, an autopsy study of elderly diabetic patients showed that diabetes was significantly associated with lacunes and increased odds of brain lacunes (*Abner et al., 2016*). A 4-year longitudinal study of cognitive function in diabetes showed a moderate cognitive decline (information-processing speed, executive functioning, and memory) in patients with diabetes (*van den Berg et al., 2010*). Other studies have shown that SVD is a crucial element in diabetic cognitive impairment (*Wallin et al., 2018*). As one of the factors of small vessel disease, lacunes may mediate the dysfunction of key brain regions. Previous studies have reported that lacunes are linked to decreases in executive function and that there is a significant positive correlation between lacunes and executive function over time (*Jokinen et al., 2011*; *Passiak et al., 2019*). The growth of lacunes is mainly located in the subcortical white matter area of the frontal lobe, which may explain the cause of impaired cognitive function, especially executive function (*Jokinen et al., 2011*). Our study also found that in patients with diabetes, lacune scores were significantly associated with impairments in cognitive function. After further excluding the influence of other factors, we evaluated the correlation between diabetes and lacunes in patients with cognitive impairment and found that the joint effect of lacunes and diabetes on cognitive impairment was stronger than the independent effects of these two factors. We speculate that the simultaneous existence of these two factors is also correlated with greater damage to cognitive function. At present, there is a lack of sufficient research to investigate the cause of this phenomenon, and we intend to study this topic further.

Lobar CMBs might also play a key role in cognitive impairment (*Yakushiji & Werring, 2016*). In this study, we also confirmed that a high score for lobar CMBs was correlated with significantly increased cognitive impairment in patients with diabetes, which may help explain the deficits in global cognitive and visuospatial executive functions observed in diabetic patients (*Yakushiji & Werring, 2016*). Our study showed no significant differences in WMH scores and PVS scores between the cognitive impairment and noncognitive impairment groups, which differs from previous research (*Passiak et al., 2019*). All subjects in the study cohort had a PVS score of 1. This situation is because even healthy people monitored the presence of PVS in this study. A study about PVS in neurologically healthy people also confirmed this phenomenon (*Yakushiji et al., 2014*). Therefore, the feasibility of evaluating cognitive impairment based on PVS score data alone needs to be further explored. Although studies have shown that PVS is associated with worse cognitive function (*Passiak et al., 2019*), this study did not find that grade of PVS is statistically significant with cognitive impairment in diabetic patients. This situation may be due to the older patients in this study. Studies have shown that age, hypertension, and genetic factors are all related to PVS (*Wardlaw et al., 2013*; *Duperron et al., 2018*; *Passiak et al., 2019*). Because research shows that age, hypertension, and genetic factors are all related to PVS, advanced age may obscure meaningful information. Expanding the sample size may be able to solve this problem. The weakened association in the clinical radiological assessment of SVD caused by the heterogeneity in the pathological substrates of WMH (*Gouw et al., 2011*) may explain why WMH had no significant statistical associations with diabetic cognitive impairment in this study.

This study still has some limitations. First, age is one of the important factors affecting cognitive function. This study found that the presence of diabetes and lacunes had a joint effect on the risk of cognitive impairment, but this may be because the overall sample in this study was older. A sample with a larger age range would be required to further understand the interaction of these two factors on cognitive impairment (such as whether there are synergistic effects in particular age groups) and to strengthen our current research conclusions. Second, due to technological limitations, we did not further study the effects of more than two risk factor interactions on cognitive function. Third, we did not have pathologic confirmation of the type of small vessel disease (SVD) in our participants. There was some information about SVD in the participants, but some vascular risk factors, including atrial fibrillation and carotid plaque, were lacking. Fourth, possible geographical biases may have resulted in a low level of education in this group of subjects. Similar to recent research (*Passiak et al., 2019*), educational background was also one of the factors associated with cardiovascular disease. Further investigation may be needed to clarify whether our findings are more generally applicable. Finally, because of the cross-sectional nature of this study, we were unable to determine the specific time of cognitive dysfunction and the progression of cognitive impairment in these patients with diabetes. Tracking the detected neuroimaging biomarkers in patients with diabetes with mild cognitive impairment might help solve this problem.

## CONCLUSION

The assessment of small vessel disease in diabetic patients with cognitive impairment found that the existence of lacunes was significantly related to cognitive dysfunction. In addition, diabetes and lacunes have a joint effect on cognitive impairment. These results suggest that monitoring lacunes, as a neuroimaging biomarker, may help to assess the deterioration in cognitive function in diabetic patients.

### Funding

This study was supported by the National Natural Science Foundation of China (61971451, 81671671); the Health and Family Planning Commission Science Foundation Projects of Hunan Province (C20180187); the Natural Science Foundation of Hunan Province (2018JJ2357); the Major Research Topics of the Health Commission of Hunan province (20201911); the Key Research and Development Projects of Hunan Province (2019SK2131); the Scientific research project of Hunan Provincial Health Commission (202109011222, 202218014518); the Nanhua University Graduate Research and Innovation Project (193YXC029); the National Training Program of Innovation and Entrepreneurship for Undergraduates (202110555106); the Key Research and Development Projects of Hunan Province (2020SK51826); and the Health and Family Planning Commission Science Foundation Projects of Hunan Province (202218014518). The funders had no role in study design, data collection and analysis, decision to publish, or preparation of the manuscript.

## Grant Disclosures
The following grant information was disclosed by the authors:
National Natural Science Foundation of China: 61971451 and 81671671.
Health and Family Planning Commission Science Foundation Projects of Hunan Province: C20180187.
Natural Science Foundation of Hunan Province: 2018JJ2357.
The Major Research Topics of the Health Commission of Hunan Province: 20201911.
The Key Research and Development Projects of Hunan Province: 2019SK2131.
Scientific research project of Hunan Provincial Health Commission: 202109011222, 202218014518.
Nanhua University Graduate Research and Innovation Project: 193YXC029.
National Training Program of Innovation and Entrepreneurship for Undergraduates: 202110555106.
The Key Research and Development Projects of Hunan Province: 2020SK51826.
Health and Family Planning Commission Science Foundation Projects of Hunan Province: 202218014518.

## Competing Interests
The authors declare that they have no competing interests.

## Author Contributions

- Hong Zhou conceived and designed the experiments, analyzed the data, authored or reviewed drafts of the paper, and approved the final draft.
- Jun Hu conceived and designed the experiments, performed the experiments, prepared figures and/or tables, and approved the final draft.
- Peihan Xie performed the experiments, prepared figures and/or tables, and approved the final draft.
- Yulan Dong performed the experiments, prepared figures and/or tables, and approved the final draft.
- Wen Chen performed the experiments, prepared figures and/or tables, and approved the final draft.
- Huiting Wu analyzed the data, prepared figures and/or tables, and approved the final draft.
- Yihong Jiang analyzed the data, prepared figures and/or tables, and approved the final draft.
- Hao Lei analyzed the data, prepared figures and/or tables, and approved the final draft.
- Guanghua Luo conceived and designed the experiments, authored or reviewed drafts of the paper, and approved the final draft.
- Jincai Liu conceived and designed the experiments, authored or reviewed drafts of the paper, and approved the final draft.

## Human Ethics
The following information was supplied relating to ethical approvals (*i.e.*, approving body and any reference numbers):

The Ethics Committee of the University of South China approved the study.

## Data Availability

The raw measurements are available in the Supplemental Files.

## Supplemental Information

Supplemental information for this article can be found online at http://dx.doi.org/10.7717/peerj.13069#supplemental-information.

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
