# Peer review of "Lacunes and type 2 diabetes mellitus have a joint effect on cognitive impairment: a retrospective study"

_PeerJ, doi:10.7717/peerj.13069_

## Round 0.1 · original submission · Major Revisions

All reviewers raised major concerns. The authors should carefully address them.

Reviewer 1 ·

Basic reporting

1. There are unclear descriptions in the manuscript. For instance, in the title, “…in patients with cerebral small vessel disease”; line 21, “…in patients with cognitive impairment”; line 84-85, “…in diabetic participants”. Which disorder was mainly studied? Similar issues can be found in many places in the manuscript. Please carefully rephrase these descriptions.
2. The language should be further polished by English editing service.
3. “p” and “t” should be shown in italics and some blank spaces were forgotten in the manuscript. Besides, the “p-value” applied in the manuscript were inconsistent. For example, “p-value” in the table 1-3, different from “p value” in the rest of the tables. Therefore, the manuscript needs another run of proofreading for typos and style.
4. Titles of all tables should be shown in formal format. The authors need to provide abbreviation in Figure 1 and all tables.
5. Some commas in Table 6 and parentheses in Table 4 were forgotten by the authors.
6. There are some grammatical errors, please have the paper revised by a native speaker.
7. In the References section, some journal names are in abbreviated forms, while others are not.
8. When abbreviation was used initially, it should follow the full word in parentheses, for instance, SVD.

Experimental design

1. In the first paragraph, the authors enumerated the association between type 2 Diabetes mellitus (T2DM) and cognitive impairment, diabetes and CSVD. The relationships among diabetes, cerebral small vessel disease, and cognitive impairment were not clearly clarified in this part. For what I know, diabetes mellitus increases the risk of cognitive decline (Ott, A., et al. 1999); T2DM may be causally associated with CSVD, in particular with lacunar stroke (Liu, J., et al. 2018); CSVD is a leading cause of cognitive decline and functional loss in the elderly (Imamine, R., et al. 2011). The authors should clarify logic among these disease types follow a “T2DM-to-cognitive impairment, possible factors-to-CSVD, and joint effect of CSVD-related biomarkers and diabetes on cognition” order in detail.
2. The authors should clearly specify their hypothesis based on previous studies in the second paragraph.
3. Most of the advances in the diagnosis, pathogenesis, epidemiology, and overall understanding of CSVD in the last decade have resulted from the clinical and research application of brain imaging, in particular MRI (Wallin et al., 2018). However, the authors need to provide truthful evidence to support the statement “Magnetic resonance imaging (MRI) is the best approach for detecting CSVD” (Line 65).
4. Please provide detail information about the inclusion and exclusion criteria for CSVD. Moreover, did all participants underwent MRI and cognitive assessment met CSVD criteria on MRI?
5. How many observers performed MRI analysis? How to evaluate interrater reliability? Please add this in the Materials and Methods section.
6. A total of 227 subjects were included in the manuscript while 241 in the “Raw_Data_2021.09.03”. Please the authors explain it.

Validity of the findings

1. How to draw the conclusion “the joint effect of diabetes and lacunes on cognitive impairment tended to be larger than the sum of the independent effects” while only patients with T2DM were included.
2. The result indicates in patients with diabetes, cognitive impairment was significantly associated with the presence of lacunes. However, there is insufficient evidence to indicate the statement “the presence of lacunes is a characteristic of small vessel disease in diabetic patients with cognitive impairment” (lines 218-220). Please the authors explain it.
3. In lines 231-233, “Consistent with the results of our study, …diabetes is significantly associated with lacunes and increased odds of brain lacunes”. Do the results include the statement in the manuscript? Please check Results section in the manuscript.
4. Please provide evidence in support of the claim “…even healthy people without risk factors can have PVS” (lines 257-258).

Additional comments

The authors carried out an interesting study to explore the joint effect of cerebral small vessel disease (CSVD)-related imaging biomarkers and diabetes on the assessment of cognitive impairment. Results of this manuscript indicated that the joint effect of diabetes and lacunas on cognitive impairment tended to be larger than the sum of the independent effects. Nevertheless, the authors need to improve their manuscript specifically with respect to their research background and the description of their findings. Below are some specific suggestions for strengthening the manuscript.

Annotated reviews are not available for download in order to protect the identity of reviewers who chose to remain anonymous.

·

Basic reporting

The authors evaluates the joint effects of diabetes and cerebral small vessel disease (CSVD)-related imaging biomarkers in patients with cognitive impairment. They found that there was a significant correlation between diabetes and cognitive impairment, and so on. The author's English writing is good, but MRI results lack pathological results. I have got several comments and questions which are included below.

Experimental design

1.To evaluate the joint effects of diabetes and cerebral small vessel disease (CSVD)- related imaging biomarkers in patients with cognitive impairment.
2.There are too many abbreviations in the abstract. It is recommended not to define abbreviations that occur less than three times.
3.There are other small problems with abbreviations, too. For example, “There was a significant correlation between diabetes and cognitive impairment (p< 0.001, X2 test). In patients with diabetes, cognitive impairment (CI) was significantly associated 34with both the presence of lacunes (p< 0.01, X2 test) and increased total SVD burden scores 35(p≤0.001, X2 test). ” Please check the abbreviations throughout the manuscript and define the abbreviations correctly.
4.Brain imaging (MRI), Brain imaging is not MRI.
5.This study adopted the use of high-resolution MRI (HR-MRI) to scan participants in our hospital for analysis. (HR-MRI) is not necessary. and other.

6.Line 83-85 “The aim of this study was to explore cognitive-related brain changes on
7.84MRI in diabetic participants and investigate whether the SVD score can be used to predict the 85occurrence of cognitive impairment in diabetic participants” but in line 20-21 (abstract) the objective is “To evaluate the joint effects of diabetes and cerebral small vessel disease (CSVD)- 21related imaging biomarkers in patients with cognitive impairment.” And I was very confused, this study was focused on patients with cognitive impairment, diabetic patients or diabetic patients with cognitive impairment? It is suggested that the author specify the main study object, and unified throughout the manuscript.
8.The inclusion and exclusion criteria for patients and controls were unclear.
9.Cognitive dysfunction not only aggravates the deterioration of the patient's physical condition but can also cause economic harm (Nasreddine et al., 2005). The reference here (Nasreddine ZS, Phillips NA, Bédirian V, Charbonneau S, Whitehead V, Collin I, Cummings JL,357 Chertkow H. 2005. The Montreal Cognitive Assessment, MoCA: a brief screening tool for mild cognitive 358 impairment. Journal of the American Geriatrics Society 53 (4):695–699. ) about the damage and damage caused by Cognitive dysfunction to patients and society is too vague and unresponsive. The reference here is a wrong reference. Please check all references and read the abstract or even the full text of the reference to ensure the correct reference.
10.Magnetic resonance imaging (MRI) is the best approach for detecting CSVD, but also works as a highly informative neuroimaging biomarker in neuroscience research (Wallin et al., 2018). This is not a proper, correct sentence.
11.What each paragraph expresses in the abstract is not clear enough. It is suggested to re-paragraph to make the content of each paragraph more coherent and logical.
12.The content of the introduction is too little, lack of summary of previous relevant research results, and the content of the proposal and assumption of the research purpose is less and not clear enough. It is suggested that the author add several summaries of relevant research results to provide more details.
13.In Figure 1, the patient with TIA OR CI, please check if there is any error here. The author did not mention TIA or CI in the “study design and subjects”.
14.This study adopted the use of high-resolution MRI (HR-MRI) to scan participants in our hospital for analysis. Please specify the patient.

15.Line 163, All subjects were divided into two groups using MMSE-2 and MOCA scores. But the author does not make it clear how to group patients.
16.Please modify the table notes of all tables to ensure better specification.
17.Line 214-215. “In this study, we used MRIs to demonstrate changes in small blood vessel damage in diabetic patients with cognitive impairment. An HR-MRI technique was used to evaluate the SVD score of patients with diabetes and cognitive impairment.” diabetic patients with cognitive impairment and patients with diabetes and cognitive impairment, Please be consistent.
18.The authors need a thorough review of abnormal functional connectivity and topological indicators in patients with diabetes.

Validity of the findings

see Q2.

Additional comments

see Q2.

---

## Round 0.2 · Minor Revisions

There are several remaining concerns that should be carefully addressed.

Reviewer 1 ·

Basic reporting

1. Did the authors let the paper be revised by a native speaker? Pleases provide the editing certificate.

Experimental design

1. In lines 90-93, “2. met the diagnostic criteria for diabetes (Harreiter & Roden, 92 2019a)”. However, how to ensure participants involved only included the patients with type 2 diabetes mellitus. Please provide the inclusion criteria in detail.
2. Some blank spaces were still forgotten by the authors. For instance, “…with cognitive impairment (p≤0.001, χ2 test)” in line 213, “(n=17 )” and “Age(median)” in Table 4 and “95%CI” in Table 6. In addition, in the description “2/t” in Table 3 and 4, “2” should be shown using superscript like “2/t”.

Validity of the findings

1.In the revised Table 1, there are some confusing expressions. First, there are significant differences between CI and non-CI subjects of the diabetic group measured using both the MMSE-2 (p < 0.001) and MoCA (p = 0.033). However, in the footnotes, “There were statistical differences in the MMSE-2 assessment in the diabetic group, so the MMSE-2 was selected as a scale for cognitive assessment before subsequent analysis.” Please the authors explain these inconsistent statements. Second, in the footnotes, the claim “a non-parametric t-test was selected” was not proper. Please clarify the statistical method used correctly. Third, how to calculate percentages in each group? Please provide these calculation formulas. Fourth, please double-check the values of two columns “p”.
2. According to flowchart of the whole study (Figure 1), a total of 218 subjects, rather than 227 subjects, were included finally for further analysis. Please the authors explain it.
3. In lines 215-216, “Table 4 shows statistics on nondiabetic patients in the entire cohort and shows that increased age (p < 0.001, χ2 test) was associated with cognitive impairment”. Nevertheless, no significant association between increased age and cognitive impairment was observed in the revised Table 4 (p = 0.438). Therefore, this seems a wrong statement based on the Table 4.

Additional comments

I am so glad to find the amendment of the manuscript made by authors. I believe it improved the manuscript substantially. However, there are still some issues existed. Thus, the authors should fix them before publication.

Annotated reviews are not available for download in order to protect the identity of reviewers who chose to remain anonymous.

·

Basic reporting

1.The author has revised the questions I raised, but the following details still need to be revised.
2.Line 24-25: Cognitive function assessments were made using the Mini-Mental State Examination (version 2), the made should be evaluated or assessed.
3.Are there too many words in the abstract? Please modify according to the requirements of the journal.
4.This study is a retrospective cohort study. A total of 227 participants (115 patients with T2DM) were enrolled in this study. It is suggested to add the information of the control group in here.
5.It is suggested that the key words imaging biomarkers should be modified to magnetic resonance.

Experimental design

6.Moreover, there is increasing evidence that the existence of CSVD contributes to cognitive impairment in patients with diabetes (Wallin et al., 2018b, 2018a).
7.A brief introduction of previous relevant studies will give us a deeper understanding of the research background. It is suggested to add 1-2 summaries of relevant studies.
8.Please state your hypothesis and object in detail in the last paragraph of the introduction.
9.2.1 Study design and subjects. Is Study design written in materials and methods appropriate? As far as I know, the design and purpose of the experiment should be presented in the last paragraph of the introduction.
10.Inclusion in criteria : 2. met the diagnostic criteria for diabetes (Harreiter & Roden, 2019a);
11.but in Abstract “A total of 227 participants (115 patients with 24T2DM) were enrolled in this study” If the inclusion criteria is diabetes, why don't 227 people here have diabetes? What about the other 112 participants?
12.All subjects were divided into two groups (cognitive impairment group and noncognitive impairment group) using the MMSE-2 and MoCA scale scores. Please briefly explain what is the standard to group.

Validity of the findings

13.We have reason to believe that the presence of lacunes is an important imaging biomarker in patients with cognitive impairment. Or in diabetic patients with cognitive impairment?
14.Magnetic resonance imaging (MRI) is an effective approach for detecting CSVD and can provide a variety of imaging biomarkers in neuroscience research. I think we've seen MRI before.

Additional comments

15.These results suggest that monitoring lacunes, as a neuroimaging biomarker, may help assess the deterioration in cognitive function in diabetic patients. This sentence needs revision.

---

## Round 0.3 · Minor Revisions

The authors should carefully consider the comments of reviewers.

Reviewer 1 ·

Basic reporting

Review:
I truly appreciate the authors’ responses and the changes they conducted. I believe it improved the manuscript substantially! There are still a few minor issues:
1. The authors claimed that the final enrollment number was 227, and the manuscripts were statistically analyzed based on 227 subjects. However, in figure 1 and lines 106-108, “After excluding subjects who had missing or unsatisfactory MRI data for analysis (n = 50), missing basic clinical information (n = 17), or missing cognitive assessments (n = 194), 227 subjects were included in this study (Figure 1)”. How to conclude that 227 subjects were included according to this statement and figure 1? Please calculate it again.
2. In line 181, “…and then the appropriate t-test was used”. “t” in “t-test” should be shown in italics.
3. Table 4 showed general characteristics of non-diabetic patients in this cohort. However, Table 4 was not mentioned in the main text after revision.

Experimental design

none

Validity of the findings

none

Additional comments

none

Annotated reviews are not available for download in order to protect the identity of reviewers who chose to remain anonymous.

·

Basic reporting

Please modify the serial numbers of other tables after deleting Table 4

Experimental design

none

Validity of the findings

none

Additional comments

none

---

## Round 0.4 · accepted · Accept

The authors have addressed all my concerns.